# Protective Efficacy of the Epitope-Conjugated Antigen N-Tc52/TSkb20 in Mitigating *Trypanosoma cruzi* Infection through CD8+ T-Cells and IFNγ Responses

**DOI:** 10.3390/vaccines12060621

**Published:** 2024-06-04

**Authors:** María Elisa Vázquez, Brenda A. Zabala, Andrea C. Mesías, Lucia Biscari, Cintia D. Kaufman, Andrés Alloatti, Francesco Siano, Gianluca Picariello, Natalia S. Corbalán, Bladimiro A. Lenis, Marta A. Toscano, Cecilia M. Parodi, Cecilia M. Pérez Brandán, Leonardo Acuña

**Affiliations:** 1Unidad de Biotecnología y Protozoarios, Instituto de Patología Experimental “Dr. Miguel Ángel Basombrío”, Consejo Nacional de Investigaciones Científicas y Técnicas (CONICET), Universidad Nacional de Salta, Salta A4400, Argentina; elisa.vazquez@conicet.gov.ar (M.E.V.); b.zabala@conicet.gov.ar (B.A.Z.); amesias@conicet.gov.ar (A.C.M.); ceciliaparodi@conicet.gov.ar (C.M.P.); 2Instituto de Inmunología Clínica y Experimental de Rosario, IDICER—CONICET—UNR, Rosario 2000, Argentina; biscari@idicer-conicet.gob.ar (L.B.); kaufman@idicer-conicet.gob.ar (C.D.K.); andresalloatti@gmail.com (A.A.); 3Istituto di Scienze dell’ Alimentazione—Consiglio Nazionale delle Ricerche (CNR), 83100 Avellino, Italy; francesco.siano@isa.cnr.it (F.S.); gianluca.picariello@isa.cnr.it (G.P.); 4Facultad de Ciencias Naturales, Universidad Nacional de Salta, Salta A4400, Argentina; nscorbalan@gmail.com; 5Unidad de Conocimiento Traslacional, Hospital Arturo Oñativia, Salta A4400, Argentina; lenisbladimiro@gmail.com (B.A.L.); m.toscano@conicet.gov.ar (M.A.T.)

**Keywords:** *Trypanosoma cruzi*, multi-epitope vaccine, chimeric protein

## Abstract

Chagas disease, caused by the protozoan *Trypanosoma cruzi*, remains a major public health challenge affecting millions in Latin America and worldwide. Although significant progress has been made in vector control, no vaccine exists to prevent infection or mitigate disease pathogenesis. We developed a rationally designed chimeric protein vaccine, N-Tc52/TSkb20, incorporating immunodominant epitopes from two *T. cruzi* antigens, the amino-terminal portion of Tc52 and the TSkb20 epitope derived from trans-sialidase. The objectives of this study were to construct and characterize the antigen and evaluate its protective potential in an immunoprophylactic murine model of *T. cruzi* infection. The N-Tc52/TSkb20 protein was recombinantly expressed in *E. coli* and its identity was confirmed using mass spectrometry and Western blotting. Immunization with the chimeric protein significantly controlled parasitemia and reduced the heart, colon, and skeletal muscle parasite burdens compared to non-vaccinated mice. Protection was superior to vaccination with the individual parental antigen components. Mechanistically, the vaccine induced potent CD8+ T-cell and IFNγ responses against the incorporated epitopes and a protective IgG antibody profile. A relatively low IL-10 response favored early parasite control. These results validate the promising multi-epitope approach and support the continued development of this type of rational vaccine design strategy against Chagas disease.

## 1. Introduction

Chagas disease, caused by the intracellular protozoan *Trypanosoma cruzi*, is a major public health challenge not only in Latin America, but also globally. It is estimated that over 6–7 million people are infected worldwide, mainly through vectorial transmission or transfusions [1,2]. Although the acute phase is often asymptomatic, decades after the initial infection, some infected individuals develop life-threatening chronic clinical manifestations, such as Chagas cardiomyopathy or intestinal disorders [3,4]. Chagas cardiomyopathy has emerged as the leading cause of parasitic heart disease in resource-poor settings, resulting in substantial mortality and disability in young adults [5].

Despite over a century of research, developing an effective vaccine against Chagas disease has proven to be challenging. Although vaccines have successfully reduced the disease burden for many infectious diseases, no vaccine is currently available to prevent or treat Chagas disease [6]. Several candidate antigens showing partial protection in animal models have been identified; however, achieving highly effective sterilization protection to prevent infection remains elusive. This challenge is compounded by the complexity of *T. cruzi* biology and lifecycle, which involves both insect vectors and mammalian hosts and persistence of intracellular forms within infected individuals [7]. The lifecycle, along with the parasite’s advanced immune evasion strategies, has significantly complicated conventional vaccine design approaches [8,9].

We have long been working on vaccine trials against *T. cruzi* in animal models, mainly using live-attenuated organisms [10,11,12]. However, these approaches come with common questions associated with live vaccines, such as production and mass distribution challenges and especially safety issues. In contrast, recombinant proteins or protein subunits are particularly attractive, as they allow the evaluation of well-defined antigens and avoid the safety limitations posed by vaccines based on live-attenuated organisms. The promising vaccine candidates evaluated to date have provided only partial or non-sterile protection in preclinical models. New platforms that explore enhanced immunogenicity and safety profiles offer opportunities to overcome this hurdle [13]. Several proteins have shown potential as antigen candidates for vaccine development, including Tc52 and Trans-sialidase (TS) [14,15,16,17]. Tc52 is an essential protein expressed across all *T. cruzi* life stages and is critical for parasite survival [18]. Tc52 is a protein that possesses glutathione transferase activity [19], and its sequence is highly conserved among strains [20]. Studies utilizing the native and recombinant forms of Tc52 reported immunogenicity, with vaccination inducing partial protection associated with protective antibodies against epitopes in the amino-terminal region [14,15]. This protective effect was comparable to that of immunization with full-length Tc52, highlighting the immunodominant nature of the N-terminal domain [15,21,22]. The trans-sialidase gene family is the largest in *T. cruzi*, with some members encoding an active TS enzyme that possesses sialic acid transferase activity. This enzyme has been shown to be crucial for parasite survival and the establishment of infection [23,24]. TS proteins are major surface and released molecules expressed at all life stages of *T. cruzi*, including both extracellular trypomastigotes and intracellular amastigotes [25]. The TS gene family expresses the highly immunodominant TSkb20 CD8+ T-cell epitope, which is targeted by approximately 30% of CD8+ T-cells during acute infection [25]. Although differences in TS gene expression between *T. cruzi* strains can lead to variation in the dominant CD8+ T-cell epitopes recognized, the TSkb20 epitope elicits CD8+ T-cell responses that show significant conservation across different DTUs of *T. cruzi* [26]. This degree of immunodominance is notable given that the CL Brenner reference genome annotates over 1400 TS family gene members, out of a total of more than 12,000 annotated genes [27].

While promising individually, vaccines based on single antigens often elicit only partial protection against *T. cruzi* antigenic diversity and sophisticated immune evasion capabilities. A multi-epitope chimeric vaccine approach aims to overcome these challenges by combining immunogenic domains from different antigens, potentially eliciting a stronger protective immunity. Several studies have emphasized the potential of designing chimeric proteins in multi-component vaccines. This approach can both reduce manufacturing and scalability burdens [28] and enhance the magnitude and quality of immune responses elicited by each component [29]. These chimeric protein vaccines have been shown to produce robust immune responses. Reports describe vaccines utilizing chimeric proteins containing multiple epitopes, such as TRASP (ASP-2/Trans-sialidase) and Traspain (N-terminal domain of Cruzipain, an α-helix linker from iTS and the central region of ASP2) against *T. cruzi*, chimT (a polypeptide based on T-cell epitopes) against *Leishmania*, or TryPan (a polypeptide based on conserved CD8+ T-cell epitopes) against *Leishmania* and *T. cruzi* [29,30,31,32]. In this study, we generated a chimeric vaccine named N-Tc52/TSkb20, incorporating N-Tc52 and two copies of the TSkb20 epitope. The design of the N-Tc52/TSkb20 chimeric protein aimed to leverage the strong immunogenic properties of both components. Tc52 is also expressed across all *T. cruzi* life stages and induces protective immune responses. Meanwhile, the TSkb20 epitope elicits a highly robust CD8+ T-cell response during acute infection. By fusing the N-terminal domain of Tc52 to TSkb20, the goal was to generate a vaccine candidate that could broadly target the parasite across its life cycle via Tc52, while also stimulating the strong CD8+ response via TSkb20. This rationally designed chimeric protein approach intended to induce superior protective immunity compared to either protein alone due to synergizing the immunogenic characteristics of both components. We hypothesized this rationally designed chimeric protein could enhance protective immunity over vaccination with the standalone parental antigens.

## 2. Materials and Methods

### 2.1. N-Tc52/TSkb20 Chimeric Protein Design and Genetic Manipulation

To design the N-Tc52/TSkb20 chimeric protein, the N-terminal sequence of Tc52 (N-Tc52) (TcCLB.503419.30) was amplified using two successive PCRs. The first PCR used 100 ng of total DNA from the *T. cruzi* CL Brener strain (DTU TcVI) epimastigotes as a template and primers F-chimera and R1-chimera (Table 1). DNA was purified using the phenol/chloroform extraction/ethanol precipitation method [33]. The PCR product was then re-amplified using primers F-chimera and R2-chimera (Table 1) to incorporate two tandem TSkb20-encoding sequences, optimizing codon usage in the reverse primer. The procedure is schematically shown in Figure 1. Reactions were carried out under standard conditions specified by GoTaq^®^ Taq DNA Polymerase (Promega, Madison, WI, USA) and the PCR program as previously reported [34,35]. Recognition sites for NheI and XhoI were added to the primers F-chimera and R2-chimera, respectively. The final PCR product was purified from agarose gel using an AccuPrep^®^ PCR/Gel Purification Kit (Bioneer, Daejeon, Republic of Korea), digested for 16 h at 37 °C with NheI (New England Biolabs, Ipswich, MA, USA) and XhoI (Promega, USA), and ligated into a linearized pRSET-A vector (Thermo Fisher Scientific, Waltham, MT, USA) using T4 DNA ligase (Promega) to fuse to a polyhistidine-encoding sequence (6xHis). *Escherichia coli* DH5α competent cells were transformed with the ligation mixture and selected on LB agar with 50 μg/mL ampicillin (Sigma-Aldrich, St. Louis, MO, USA) to obtain p-CHIM. Positive colonies were confirmed by PCR, p-CHIM was purified, and the construct was verified by DNA sequencing (CERELA-CONICET; Tucumán, Argentina).

### 2.2. N-Tc52/TSkb20 Chimeric Protein Expression and Purification

The purified p-CHIM plasmid was transformed into *E. coli* BL21[DE3] cells, which were then plated on LB agar containing 50 μg/mL ampicillin. Single colonies were selected and used to inoculate 75 mL of LB medium supplemented with 200 μg/mL ampicillin and 0.6% glucose. The cultures were incubated at 37 °C with shaking at 160 rpm. The overnight culture was transferred to 1.5 L LB in a 3 L baffled flask and incubated at 37 °C with shaking at 160 rpm for 2 h until mid-log phase, as assessed by measuring absorbance at 600 nm (Abs600). Protein expression was induced by adding 0.8 mM isopropyl β-D-1-thiogalactopyranoside (IPTG; Genbiotech, Buenos Aires, Argentina) and incubating for an additional 3 h. Cells were pelleted by centrifugation at 10,000× *g* for 10 min at 4 °C, washed twice with phosphate-buffered saline (PBS) pH 7.4), and lysed by repeated freeze–thaw cycles followed by sonication.

N-Tc52/TSkb20 protein was obtained as inclusion bodies (IBs). IBs were washed 3 times with 5% Triton X-100 (BioRad, Hercules, CA, USA), and once with 5% ε-poly-L-lysine. Washing with these compounds serves to solubilize contaminating proteins and remove endotoxins present [36,37,38]. IBs were then washed twice with Buffer A (BA: 50 mM Tris-HCl, 300 mM NaCl, 0.5 mM phenylmethylsulfonyl fluoride, 0.5 mM β-mercaptoethanol, pH 8) to remove detergent residues. Clean IBs containing N-Tc52/TSkb20 were solubilized by freezing/sonication/thawing cycles in the presence of 2% sodium lauroyl sarcosinate (Sarkosyl; Sigma-Aldrich). After solubilization, 5% Triton X-100 was added. Soluble chimeric protein was purified by immobilized metal affinity chromatography using nickel-nitrilotriacetic acid (Ni-NTA) resin (Roche, Mannheim, Germany) as follows: the sample was diluted with BA to 0.2% sarkosyl and 0.5% Triton X-100 and then applied to a 1 CV Ni-NTA column pre-equilibrated with BA + 0.2% sarkosyl. The column was washed with 5 CV BA + 0.2% sarkosyl + 5 mM imidazole and 5 CV BA + 0.2% sarkosyl + 10 mM imidazole. N-Tc52/TSkb20 was eluted with 10 CV BB (BA + 0.2% sarkosyl, 1% glycerol, 450 mM imidazole, pH 7). The fractions were analyzed by SDS-PAGE.

The eluate was precipitated with 85% saturated ammonium sulfate. The precipitate was resuspended in PBS (pH 7.4) containing 5% glycerol and treated with an ε-poly-L-lysine column (Pierce High-capacity Endotoxin Removal Resin, Thermo Fisher Scientific, France) following the manufacturer´s instructions to remove endotoxins potentially present. Proteins were concentrated and buffer exchanged with PBS (pH 7.4) and 1% glycerol using Amicon centrifugal filters (Merck Millipore, Burlington, MA, USA). The protein samples were centrifuged for 30 min at 12,000× *g* and filtered through 0.22 μm filters. This yielded a preparation that remained fully soluble and sterile until use. Residual endotoxins were quantified using the Limulus Amebocyte Lysate assay (LAL; Thermo Fisher Scientific, Saint-Herblain, France).

The final protein concentration was determined using the Bradford assay. Approximately 15 mg of pure chimeric protein containing less than 0.1 endotoxin unit (EU)/mg protein was obtained per liter of culture. The recombinant parental proteins rTc52 and rTS were used for comparison in vaccination assays. These proteins were obtained from inclusion bodies in our laboratory. Expression and purification methods similar to those used for N-Tc52/TSkb20 were followed, using the p-TS and p-Tc52 plasmids, which contain the structural genes encoding the TS and Tc52 proteins, respectively. The p-TS plasmid was obtained through sequential cloning of the TS-encoding gene after its amplification by PCR using the F-TS and R-TS primers (Table 1), whereas the p-Tc52 plasmid was constructed by cloning the gene after its amplification using the F-Tc52 and R-Tc52 primers (Table 1). Minor modifications were performed to obtain the highest purity of each protein. The pure and fully soluble rTc52 and rTS proteins in their final forms are shown in Appendix A.

### 2.3. Characterization of N-Tc52/TSkb20 Chimeric Protein by Liquid Chromatography–Mass Spectrometry (HPLC–MS)

Nanoflow liquid chromatography—high-resolution tandem mass spectrometry (HPLC-MS/MS) analysis was used to confirm the presence of N-Tc52/TSkb20 chimeric protein either in the ammonium sulfate pellet or in the SDS-PAGE protein band. The protein pellet was suspended in 0.4 mL of denaturing buffer (6 M guanidine, 100 mM Tris, 1 mM EDTA, pH 8.0), and cysteins were reduced with 10 mM dithiothreitol (DTT) at 55 °C for 1 h and alkylated with 55 mM iodoacetamide at 25 °C for 30 min in the dark. Proteins were desalted using Zeba^®^ size exclusion chromatography spin columns (Pierce Biotechnology, Rockford, IL, USA, 7 kDa cut-off), eluted in 50 mM ammonium bicarbonate (pH 7.8), quantified using the Bradford assay, and digested with proteomic grade trypsin (Pierce Biotechnology, Rockford, IL, USA) at a 1/50 (*w*/*w*) trypsin-to-protein ratio overnight at 37 °C. The SDS-PAGE protein band was manually excised, destained with 50% acetonitrile/50 mM ammonium bicarbonate, Cys-reduced/alkylated by sequential treatment with 10 mM DTT and 55 mM iodoacetamide and in-gel digested with 10 μL of 12.5 ng/μL proteomic grade trypsin overnight at 37 °C. Tryptic peptides were extracted in 50% acetonitrile/1% formic acid, vacuum-dried, and finally reconstituted in 0.1% formic acid.

HPLC-MS/MS analyses were performed using an Ultimate 3000 nanoflow ultrahigh-performance liquid chromatograph (Dionex/Thermo Scientific, San Jose, CA, USA) coupled with a Q Exactive Orbitrap mass spectrometer (Thermo Scientific). Tryptic peptides from batch (1 μg) or in-gel (1/10 of the pool) digestion were loaded through Acclaim PepMap 100 (75 μm i.d. × 2 cm) trap columns (Thermo Scientific) using a Famos autosampler (Thermo Scientific) and separated using an EASY-Spray PepMap C18 column (2 μm, 25 cm × 75 μm) with 3 μm particles (Thermo Scientific). Eluent A was 0.1% formic acid (*v*/*v*) in LC-MS-grade water, and eluent B was 0.1% formic acid (*v*/*v*) in 80% acetonitrile. Peptides were separated by applying a 2–45% gradient of B over 60 min after 10 min of isocratic elution at 2%, at a constant flow rate of 300 nL min^−1^. Spectra were acquired in the positive ionization mode, scanning the 300–1600 m/z range, with a resolving power of 70,000 full width at half-maximum (FWHM) for precursors and 17,500 for fragments. The spectrometer operated in top10 data-dependent acquisition, applying a 10 s dynamic exclusion. Peptide sequences were assigned by bioinformatic spectral matching using Proteome Discoverer 2.1 software (Thermo Scientific), and the predicted sequence of N-Tc52/TSkb20 and the *E. coli* protein database downloaded from UniprotKB (May 2021) with the SEQUEST algorithm. The search parameters were as follows: carbamidomethylation of cysteines as a static modification; methionine oxidation, pyroglutamic acid at N-terminus glutamine as variable modifications; mass tolerance value of 8 ppm for precursor ion and 12 ppm for MS/MS fragments; and trypsin as the proteolytic enzyme with up to two missed cleavages and semi-tryptic cleavage allowed. The confidence of identification was set at 0.1% false discovery rate Peptide Spectrum Matches (PSMs) calculated by target decoy filtering.

### 2.4. Animal Model and Immunization Protocol

To evaluate this novel N-Tc52/TSkb20 antigen as an immunogen capable of controlling *T. cruzi* infection, one-month-old female C57BL/6 mice (n = 10 per group) were used in immunization and challenge studies. Previous reports have indicated potential differences in acute *T. cruzi* infection between females and males. Specifically, females reportedly show greater resistance to infection as well as more homogeneous and reproducible results [39,40,41,42,43]. For this initial study, we selected the female mouse model to potentially allow for a better assessment of vaccine efficacy in our preliminary evaluation. Animals were housed in cages of up to five mice each under a 12 h light/dark cycle at 25 °C with free access to standard laboratory chow and water. Mice were bred, and experiments were conducted at the Animal Facility of the Instituto de Patología Experimental and Universidad Nacional de Salta, Argentina.

Mice were inoculated with three doses, three weeks apart, via the subcutaneous route, as previously reported [30,44,45,46], consisting of 50 μL containing 20 μg of chimeric protein and 15 μg of QuilA adjuvant (InvivoGen, Toulouse, France). Additional groups received injections of 20 μg rTc52 + 15 μg QuilA, 20 μg rTS + 15 μg QuilA, or QuilA (non-vaccinated). We selected the QuilA adjuvant mainly because of its ability to potentiate an immune response that is compatible with combating *T. cruzi*. QuilA is a saponin adjuvant. Saponins induce a strong adjuvant response to both T-dependent and T-independent antigens. Saponins also induce robust cytotoxic CD8+ lymphocyte responses [47,48]. Boosters were administered every 21 days in order to avoid overlap between immunization and the antibody peak around days 10–14 post-inoculation, when these could interfere with antigen action. This dosing schedule follows a commonly reported approach [49] and has been frequently utilized in prior studies from our group [44]. Twelve days after the final dose, blood was collected from the tail tips of anesthetized mice for IgG determination and half of the mice were sacrificed to obtain their spleens and measure cellular immune responses during the expansion phase. The other half of the animals were challenged with an infective dose of *T. cruzi* at thirty-five days after the last dose. 

### 2.5. T. cruzi Challenge and Parasitemia Assessment

To assess short-term protection, 35 days after the final immunization, the remaining mice were challenged intraperitoneally with 1000 bloodstream trypomastigotes from the infective Tulahuen strain (DTU VI), which was previously maintained by serial passages in C57BL/6 mice. Parasitemia was monitored by enumerating the number of parasites per 50 fields from 10 μL of fresh blood collected from the tail vein of anesthetized mice twice weekly for 25 days under a light microscope (40× magnification). The daily survival rate was monitored until day 25, at which point the experiment was terminated and all remaining animals were sacrificed after the first mouse in the non-vaccinated group died. This ensured the degree of infection could be assessed at the same time point for all animals.

### 2.6. Assessment of Tissue Parasite Burden

Twenty-five days after the challenge, mice were humanely euthanized by CO_2_ exposure, and heart, colon, and skeletal muscle samples from the rear legs were collected. Each tissue was cut in the sagittal plane and used to determine parasite burden using quantitative real-time PCR (qRT-PCR). Total DNA was isolated from 20 mg of tissue samples using an ADN-Puriprep Highway nucleic acid kit (InbioHighway, Tandil, Argentina) according to the manufacturer’s instructions. qRT-PCR was performed on a QuantStudio5 thermal cycler (Applied Biosystems, Foster City, CA, USA) with 10 μL reactions containing 40 ng of total DNA, 5 μL of iTaqTM Universal SYBR^®^ Green Supermix (BioRad, Hercules, CA, USA), and 1 μM *T. cruzi*-specific 18S oligonucleotides (F-Sat and R-Sat, Table 1 [50]). The cycling conditions were as follows: initial denaturation at 95 °C for 10 min, followed by 40 cycles of 95 °C for 15 s and 63 °C for 30 s. Data were normalized to murine TNF-α amplification [51], results are shown as the number of parasites/40 ng of DNA. For quantification, a standard curve of total parasites with serial dilutions was obtained. Data were analyzed using the QuantStudio Design & Analysis 2.5.1 Software from Applied Biosystems.

### 2.7. Histological Analysis

To determine if immunization was effective in reducing nesting and tissue damage caused by *T. cruzi*, half of the remaining heart, colon, and skeletal muscle samples were fixed in 10% buffered formalin and dehydrated through an ethanol series before embedding in paraffin. Sections (3 μm thick) were cut and stained with hematoxylin and eosin (H&E). A double-blind microscopic analysis of all tissue sections (n = 4 per group) was performed on coded slides. Using a light microscope, the number of amastigote nests was recorded and characterized as follows: - = absent, * = 1 group, ** = 2–4 groups, *** = 4–10 groups, **** = ≥10 groups. Cellular inflammation was graded as + for small inflammatory foci, ++ for medium foci, and +++ or higher for large foci. Images were acquired using a Leica ICC50W camera (Leica, Wetzlar, Germany) attached to the light microscope.

### 2.8. Detection of Specific IgG Subclasses

Serum samples collected 12 days after the final immunization were aliquoted and stored at −20 °C until use for IgG subclass determination. Ninety-six-well plates were coated overnight at 4 °C with 0.25 μg/well of chimeric protein, rTc52, or rTS in carbonate buffer (0.2 M, 0.02% sodium azide, pH 9.6). IgG subclasses (IgG1 and IgG2c) were measured using an enzyme-linked immunosorbent assay kit (Sigma-Aldrich, St. Louis, MO, USA). Plates were blocked with 5% non-fat dry milk in PBS for 1 h at 37 °C, and then incubated with serum samples (1:20 dilution, 100 μL/well) for 1 h at 37 °C. Biotin-conjugated goat anti-mouse IgG subclasses antibodies (IgG1 or IgG2c; BD Biosciences, San José, CA, USA) were added at 1:4000 and 1:2000 dilutions, respectively, and incubated for 1 h at 37 °C. Streptavidin-horseradish peroxidase conjugate (BD Biosciences) was added at a 1:7500 dilution and incubated for 1 h at 37 °C. Color development was performed using the TMB Substrate Reagent Set (BD Biosciences) for 20 min, and the reaction was stopped with 2N H_2_SO_4_. The optical density was measured using a TECAN infinite f50 microplate reader (Tecan, Männedorf, Switzerland). The antibody titers for IgG subclasses are presented as optical density (OD) values obtained from absorbance measurements at 450 nm.

### 2.9. Western Blot Characterization of N-Tc52/TSkb20 via Serum Antibody Detection

The purified N-Tc52/TSkb20 protein was separated by 12% SDS-polyacrylamide gel electrophoresis, then transferred to an Immobilon-P polyvinylidene difluoride (PVDF) membrane (Millipore, Burlington, MA, USA). The membrane was initially blocked by incubation in 5% non-fat dry milk dissolved in 50 mM Tris (pH 7.0), 150 mM NaCl buffer for 1 h at room temperature. It was then washed three times with Tris-buffered saline supplemented with 0.05% Tween 20 (TBST). Next, the membrane was incubated overnight at 4 °C with a 1:100 dilution of serum pool from N-Tc52/TSkb20-immunized mice. Following washing with TBST, horseradish peroxidase-conjugated anti-mouse secondary antibody (Sigma-Aldrich, St. Louis, MO, USA) at a 1:5000 dilution was applied for 1 h at room temperature. Immunoreactive bands were visualized using an enhanced chemiluminescent substrate and exposed on X-ray film (GE Healthcare Life Sciences, Pittsburgh, MA, USA).

### 2.10. Splenocyte Cell Culture

Spleens were aseptically removed from euthanized mice 12 days after the final immunization, macerated through sterile mesh, and resuspended in RPMI 1640 medium supplemented with L-glutamine (Biological Industries, Beit Haemek, Israel). Cells were centrifuged at 1500 rpm for 10 min at 4 °C in a fixed-angle rotor centrifuge and resuspended in lysis buffer (0.17 M Tris, 0.16 M NH4Cl, pH 7.2) to remove erythrocytes. The remaining splenocytes were washed in RPMI and resuspended in RPMI medium supplemented with 10% fetal bovine serum. Cell viability was assessed using trypan blue exclusion, and cell numbers were determined using a Neubauer hemocytometer under a light microscope.

### 2.11. Measurement of IL-10 Cytokine Response

Splenocyte culture supernatants at a concentration of 2 × 10^6^ live cells/mL, collected after 48 h of stimulation with 0.2 μM N-Tc52/TSkb20 protein or 15 μg/mL of total soluble proteins homogenate from the Tulahuen *T. cruzi* strain (HP-TUL) were used to measure IL-10 levels via ELISA (optEIA ELISA kits; BD Biosciences, San José, CA, USA), according to the manufacturer’s instructions. Briefly, 96-well plates were coated overnight at room temperature with a 1:250 diluted Capture Antibody in carbonate buffer (0.2 M, 0.02% sodium azide, pH 9.6). The plates were blocked for 1 h at room temperature with 10% fetal bovine serum in PBS, and then incubated for 2 h at room temperature with 100 μL/well of supernatant samples. A standard curve was prepared according to the manufacturer’s guidelines. The detection Antibody (1:250 dilution) and streptavidin-horseradish peroxidase conjugate (1:250 dilution) were simultaneously incubated for 1 h at room temperature. Color development was performed using the TMB Substrate Reagent Set (BD Biosciences) for 30 min and the reaction was stopped with 2N H_2_SO_4_. The optical density was measured at 450 nm using a TECAN infinite f50 microplate reader (Tecan, Männedorf, Switzerland). Measurements were interpolated against each standard curve.

### 2.12. ELISPOT Assay for IFNγ Determination

An ELISPOT assay (BD Biosciences, San José, CA, USA) was performed according to the manufacturer’s recommendations to determine whether splenocytes from N-Tc52/TSkb20-immunized mice produced IFN-γ upon restimulation with N-Tc52/TSkb20 protein and TSkb20 peptide. Briefly, 96-well plates were coated with IFNγ antibody, blocked, and 1 × 10^6^ cells/well were cultured in plates and restimulated with 0.4 μM TSkb20 peptide or 0.2 μM N-Tc52/TSkb20 protein (epitope equimolar concentration), at 37 °C, 5% CO_2_ for 24 h. After incubation and cell removal, the wells were incubated with biotinylated detection antibody, and streptavidin-alkaline phosphatase conjugate was developed with 3,3′-diaminobenzidine (DAB) substrate (Sigma-Aldrich) and hydrogen peroxide. Spots were counted using an automated ImmunoSpot S5.0.3 ELISPOT reader and analysis software (Instituto Nacional de Parasitología Dr. Mario Fatala Chaben, Buenos Aires, Argentina).

### 2.13. Activation-Induced Marker (AIM) Assay

Suspended splenocytes were also used to measure the activation of specific CD8+ T-cells after restimulation with the N-Tc52/TSkb20 chimeric protein and TSkb20 peptide. An activation-induced marker (AIM) assay was performed to assess the expression of the early activation markers CD69 and CD25 on CD8+ T-cells. Briefly, 1 × 10^6^ cells/well were cultured in U-bottom 96-well plates and restimulated with 0.4 μM TSkb20 peptide or 0.2 μM N-Tc52/TSkb20 protein, at 37 °C, 5% CO_2_ for 15 h. Unstimulated and 5 μg/mL concanavalin A (ConA)-stimulated cells served as controls at a final volume of 200 μL. After incubation, cells were stained using flow cytometry. Cell suspensions were stained with anti-MHCII-APCCy7, anti-B220-APCCy7, and LIVE/DEADTM near-IR dye (exclusion panel or DUMP channel), followed by anti-TCRβ-PE, anti-CD8-PerCP, anti-CD4-FITC, anti-CD25-APC, and anti-CD69-PECy7 to analyze the activation of CD8+ T-cells. Cells were fixed in 4% formaldehyde before acquisition on a BD FACS Canto II flow cytometer (BD Biosciences). Data were analyzed using FlowJo vX.0.7 software.

### 2.14. In-Silico Prediction of CD8+ T-Cell and B-Cell Potential Epitopes

MHC-I binding affinity predictions were performed using the computational prediction software NetMHCpan version 4.1b by applying the following parameters: peptide length, 8–11; rank threshold for strong binder, <0.5; and rank threshold for weak binder, <2. Peptides were evaluated only for the C57BL/6 mouse H2-Kb allele. Only epitopes with a high binding affinity, which could have immunogenic potential, were considered.

B-cell epitopes were predicted only as linear epitopes using BepiPred Linear Epitope Prediction 2.0 [52]. This module predicts the continuous amino acid stretches which can act as a potential peptide to bind antibodies based on the physicochemical properties of amino acids (hydrophilicity, flexibility, accessibility, turns and exposed surface).

### 2.15. Statistical Analysis

All datasets were subjected to a one-way analysis of variance (ANOVA) followed by Tukey’s post hoc honestly significant difference test for inter-group comparisons. Data are expressed as the mean ± standard error of the mean (SEM) from at least three independent experiments unless otherwise stated. Statistical analysis was conducted using Statistix software version 9.0. Statistical significance was established as *p* < 0.05.

## 3. Results

### 3.1. Development and Characterization of the N-Tc52/TSkb20 Chimeric Protein Construct

The chimeric construct encoding N-Tc52/TSkb20 was successfully cloned into the pRSET-A plasmid vector and designated p-CHIM (Figure 1). Next, the N-Tc52/TSkb20 chimeric protein was satisfactorily expressed in *Escherichia coli* BL21 [DE3]. As detailed in Section 2.2, after solubilization of the inclusion bodies, the purified protein was obtained in a soluble form using affinity chromatography. Finally, residual lipopolysaccharides (LPS) were removed. The N-Tc52/TSkb20 protein was further ultrafiltered to exchange the buffer with an adequate buffer composition for in vivo experiments. This resulted in a pure, fully soluble chimeric protein comprising 253 amino acid residues in its final form (Figure 2A).

The effective expression of N-Tc52/TSkb20 was confirmed using HPLC-MS/MS-based sequencing of tryptic peptides. Bioinformatic-assisted sequencing of peptides clearly demonstrated the presence of N-Tc52/TSkb20, which was inferred from 122 Peptide Spectrum Matches (PSMs) covering 86.6% of the protein sequence, including residue no. 3 and residue no. 248 (out of 253) at the N- and C-termini, respectively. Several semi-tryptic peptides were identified alongside the expected tryptic peptides, while some matching peptides occurred in duplicated forms with reduced or oxidized methionine. Together with the target protein that occurred at relatively high abundance, as estimated by peptide ion intensity, 74 low-abundance *E. coli* gene products co-precipitating from the expression medium were identified with high confidence (1% FDR, >2 PSMs). Analysis of the in-gel digested protein band that migrated at an estimated MW of 29.4 kDa confirmed that the purified protein was N-Tc52/TSkb20, inferred from 76 PSMs covering 75.5% of the protein sequence. N-Tc52/TSkb20 was by far the dominant protein band in the gel demonstrating that it was enriched in the ammonium sulfate precipitate.

### 3.2. N-Tc52/TSkb20 Immunization Effectively Controls Parasitemia and Reduces Tissue Parasite Burden in a Mouse Model of T. cruzi Infection

Subsequently, we tested the protective potential of N-Tc52/TSkb20 against *T. cruzi* infection using a mouse model. C57BL/6 mice were administered N-Tc52/TSkb20 with the QuilA adjuvant, as shown in Figure 3A. After the final dose at 35 days, animals were inoculated with an infective dose of *T. cruzi* bloodstream trypomastigotes from the Tulahuen strain. As shown in Figure 3B, parasite levels in the peripheral blood were significantly lower in N-Tc52/TSkb20-immunized animals compared than in the non-vaccinated group at 25 dpi (*p* < 0.01). Although the parasitemia kinetic profile of mice immunized with the N-Tc52/TSkb20 chimera was similar to that of the rTc52 and rTS parental proteins, early parasite control was better achieved and regulated over time in the N-Tc52/TSkb20-immunized group. This result was more evident when observing the cumulative parasitemia in the area under the curve graph (Figure 3C). We found that N-Tc52/TSkb20 immunization was better able to control parasitemia in a faster, more sustained and consistent manner than its parental proteins (*p* < 0.05 vs. rTc52 and rTS). In addition, daily recordings of animal survival were made until sacrifice on day 25, which marked the humanitarian endpoint after the death of the first mouse.

We next investigated whether the decreased parasite burden in the blood was also reflected in the principal target organs of *T. cruzi* infection: colon, heart, and skeletal muscle. These organs were removed at 25 dpi at the end of the experiment, and qRT-PCR was performed to quantify the *T. cruzi* satellite DNA.

First, as illustrated in Figure 4, the parasite burden in all three organs was significantly lower in the N-Tc52/TSkb20-immunized group than in the non-immunized animals (*p* < 0.001 vs. non-vaccinated). In fact, *T. cruzi* DNA levels in the N-Tc52/TSkb20-immunized group were less than 90% in the hearts (*p* < 0.001) and 75% less in the colon and muscle (*p* < 0.001) than in the non-vaccinated group. Comparing samples from the N-Tc52/TSkb20 with rTc52- and rTS-immunized groups, no differences were found in the colon. However, heart samples from N-Tc52/TSkb20 immunization showed a considerably diminished parasite burden compared to rTc52 (*p* < 0.05) (Figure 4B). The muscle was the tissue where the highest amount of DNA parasite was detected. Additionally, in these samples, the main differences in the DNA parasite load were observed among the experimental groups, as shown in Figure 4C. When compared to its parental antigens rTc52 and rTS, the N-Tc52/TSkb20-immunized group was nearly 50% more effective at controlling parasite burden in the muscle (*p* < 0.05 vs. rTc52 and rTS).

To further characterize the parasite burden, we quantified amastigote nests in skeletal muscle tissues by microscopic examination. The quantity of amastigote nests and degree of inflammation were categorized into a range, as shown in Figure 5B. The non-immunized group contained 6–10 parasite foci per sample. No intracellular parasites were found in the muscle samples of N-Tc52/TSkb20-immunized animals. In the groups immunized with rTc52 and rTS, 2–4 amastigote nests were found in each animal. Regarding inflammation, the muscle of N-Tc52/TSkb20-immunized animals showed minimal cellular infiltration (mostly mononuclear cells) compared to non-immunized and rTc52- or rTS-immunized animals. Tissues from non-infected control animals were used as baseline. Representative images of inflammation and the presence of amastigote nests are shown for each experimental group (Figure 5A). NTc52/TSkb20 chimeric immunization reduced the number of intracellular parasite nests and the associated inflammation of acute infection, correlating with attenuated tissue damage in skeletal muscle at 25 dpi, as demonstrated by scoring in Figure 5B.

### 3.3. N-Tc52/TSkb20 Immunization Induces a Moderate but Protective Anti- N-Tc52/TSkb20 Humoral Immune Response in a Mouse Model of T. cruzi Infection

We sought to further characterize the N-Tc52/TSkb20 immune correlates. As antibodies and B-cell responses play an important role in defense against extracellular forms of parasites, we aimed to analyze the humoral immune response induced by the N-Tc52/TSkb20 chimeric vaccine in our next series of experiments. Western blot analysis was performed to characterize both the humoral immune response as well as the size and integrity of the recombinant N-Tc52/TSkb20 chimeric protein. Serum samples from mice vaccinated with N-Tc52/TSkb20 clearly detected a single, discrete band on immunoblots of the purified antigen, demonstrating the production of antibodies targeting the recombinant antigen in immunized mice (Figure 6A). Additionally, this band was present at the expected molecular weight, confirming the presence and integrity of the full-length protein. These results establish that vaccination elicited a specific humoral response and validate the anticipated structure and immunogenic epitopes of the N-Tc52/TSkb20 construct. To further explore the humoral immune response, we investigated the specific humoral response by measuring the IgG antibody subclasses. Mice vaccinated with N-Tc52/TSkb20 demonstrated detectable anti-chimeric IgG2c antibody levels against N-Tc52/TSkb20, in contrast to non-detectable IgG1 levels within the same vaccinated mice. This resulted in a notable predominance of IgG2c over IgG1, as determined by comparing the mean antibody levels between the two IgG subclasses among immunized animals (Figure 6B), indicative of a Th1-biased antibody response. Non-vaccinated mice, as expected, did not exhibit significant levels of either IgG subclass. The antibodies produced were specific for N-Tc52/TSkb20 and did not recognize the rTc52 or rTS proteins. Compared to mice immunized with the parental proteins, only one mouse immunized with rTc52 produced IgG2c antibodies that recognized both rTc52 and N-Tc52/TSkb20. In the rTS-immunized group, no mouse elicited anti-TS or anti-N-Tc52/TSkb20 antibodies.

### 3.4. Elucidating the Combined CD8+, IFNγ and Regulatory Cytokine Facets of the Immune Response Induced by the N-Tc52/TSkb20 Vaccine

Our previous findings have demonstrated that the N-Tc52/TSkb20 vaccine elicits a humoral response. Given the crucial role of cell-mediated defenses in controlling the intracellular parasite burden, we sought to characterize the key aspects of the vaccine-induced cellular response. IL-10 is a regulatory cytokine that can dampen protective immunity [53,54]. Therefore, we first measured its secretion by splenocytes from immunized mice. Splenocytes were isolated 12 days after the final immunization and stimulated ex vivo with the immunizing antigen N-Tc52/TSkb20 or soluble *T. cruzi* total extract (HP-TUL). Remarkably, N-Tc52/TSkb20 immunization significantly reduced IL-10 levels in the supernatants compared to those in the non-vaccinated group (*p* < 0.001; Figure 7A). These levels were nearly undetectable after stimulation. Additionally, N-Tc52/TSkb20 immunization decreased IL-10 levels more effectively than immunization with parental rTc52 and rTS proteins. Specifically, a greater reduction was observed in the N-Tc52/TSkb20 group than in the rTc52 and rTS groups when cells were stimulated with either N-Tc52/TSkb20 (*p* < 0.05 vs. both) or HP-TUL (*p* < 0.01 vs. both). These findings indicate that the N-Tc52/TSkb20 vaccine is superior in downregulating the cytokine IL-10 upon antigen re-stimulation of splenocytes. Given IL-10’s ability to dampen protective immune responses, these results suggest that the N-Tc52/TSkb20 vaccine may promote a more effective anti-parasitic immune response by decreasing IL-10 secretion.

To further characterize the cellular immune response induced by N-Tc52/TSkb20, we evaluated CD8+ T-cell priming, given their critical role in host resistance to *T. cruzi* and considering their important role in defense against the parasite. Therefore, we assessed vaccine-induced CD8+ T-cell responses and IFNγ production using multiparameter flow cytometry and ELISPOT assay, respectively. Given the design of N-Tc52/TSkb20 to incorporate two CD8 epitopes from Trans-sialidase, we hypothesized that this vaccine could elicit robust CD8+ T-cell immunity critical for protection.

We further investigated the cellular responses and performed an ELISPOT assay to measure IFNγ secretion upon restimulation. IFNγ is an important cytokine that controls *T. cruzi* infection. Figure 7B (top) shows representative images of spot formation in the ELISPOT assay from the different experimental groups. Spots indicating IFNγ secretion were observed in the N-Tc52/TSkb20-immunized group after stimulation with the TSkb20 peptide or N-Tc52/TSkb20 chimeric protein. We showed that spleen cells from mice immunized with the N-Tc52/TSkb20 protein secreted significantly more IFNγ compared to non-immunized mice (*p* < 0.001) upon stimulation with TSkb20 peptide. Importantly, there were also significant differences (*p* < 0.05) in IFNγ-secreting cells between the N-Tc52/TSkb20-immunized group compared to the non-immunized mice after stimulation with the N-Tc52/TSkb20 protein (Figure 7B). To examine the specific activation of CD8+ T-cells, we performed an activation-induced marker (AIM) assay as described previously [41]. Splenocytes were incubated for 14 h after restimulation with N-Tc52/TSkb20 protein or TSkb20 peptide. We then determined the percentage of CD8+ and CD4+ cells expressing the activation markers CD25 and CD69 using flow cytometry following the gating strategy shown in Figure 7C. Restimulation with the TSkb20 peptide induced a significant increase in CD8+ CD69+ CD25+ cells in N-Tc52/TSkb20-immunized mice compared to controls (*p* < 0.01), while no significant increase was detected after N-Tc52/TSkb20 protein stimulation (Figure 7C). We also did not observe any differences in the CD4+ T-cell populations, whether expressing or not the activation markers CD25 and CD69, among groups stimulated with either N-Tc52/TSkb20 or TSkb20 alone. The results of the AIM assay show that the TSkb20 peptide activates CD8+ T-cells in mice immunized with the N-Tc52/TSkb20 vaccine, confirming the critical role of TSkb20 epitopes in triggering the CD8+ cellular immune response, without significant contribution from CD4+ populations.

### 3.5. In-Silico Prediction of Novel CD8+ T-Cell and B-Cell Epitopes within the N-Tc52/TSkb20 Chimeric Vaccine Correlated with Its Enhanced Protective Efficacy

Given the superior protective efficacy afforded by immunization with the N-Tc52/TSkb20 vaccine relative to rTc52 and rTS, we theorized that one of the possible differences in the immune response could stem from epitopes emerging from chimeric conformations that are not present individually. CD8+ T-cells play an important role in protective immunity against intracellular parasites, such as *T. cruzi*, through the recognition of pathogen peptides presented by MHC class I molecules [55]. Using NetMHCpan 4.1, we performed in silico prediction of potential peptide epitopes that could bind strongly to the MHC class I allele H2-Kb and thus stimulate protective CD8+ T-cell responses within the N-Tc52/TSkb20 vaccine candidate. We identified five novel peptides with high predicted binding affinities for H2-Kb (Table 2). These novel potential epitopes arose from the fusion of N-Tc52 and TSkb20 sequences within the chimeric protein vaccine. Specifically, the peptides resulted from the fusion of (i) The C-terminal portion of N-Tc52 with the N-terminal TSkb20 (SANYKFTL and SANYKFTLV) and (ii) The two TSkb20 sequences (ANYKFTLVA, VANYKFTL, and VANYKFTLV). Importantly, the novel epitopes bear similarity to the known TSkb20 epitope (ANYKFTLV) and conserve their hydrophobicity.

Analysis with BepiPred Linear Epitope Prediction 2.0 to predict B cell epitopes identified five epitopes mapped to Tc52. One novel potential epitope (PKSHVTWSANYK) was predicted to not correspond to sequences in individual proteins (Table 2). This epitope is likely generated through the fusion of N-Tc52 with the first TSkb20 sequence. Its novel composition provides evidence that conjugation generates a distinct immunogenic surface compared with separate proteins.

## 4. Discussion

The current study aimed to develop and characterize a novel chimeric protein vaccine containing Tc52 and TSkb20 epitopes from *Trypanosoma cruzi*, and to evaluate its protective potential in a murine model of *T. cruzi* infection. We successfully generated a N-Tc52/TSkb20 chimeric antigen and confirmed its expression and structure using genetic, biochemical, and mass spectrometry-based analyses. Immunization with this antigen effectively controlled parasitemia and significantly reduced the tissue parasite burden in the heart, colon, and skeletal muscle of infected mice compared to non-vaccinated controls. Moreover, N-Tc52/TSkb20 elicited stronger protection than the parental proteins rTc52 and rTS, demonstrating the enhanced immunogenic properties of the designed chimeric antigen. Collectively, these results showed that the objectives of generating and characterizing N-Tc52/TSkb20, as well as assessing its protective efficacy in vivo, were successfully achieved in this study.

### 4.1. Generation and characterization of the N-Tc52/TSkb20 chimera

The N-Tc52/TSkb20 recombinant antigen was generated using genetic engineering. Successful generation of the chimeric construct was confirmed by sequencing of the resultant genetic fusion. The N-Tc52/TSkb20 protein was expressed in *E. coli* and purified to homogeneity from inclusion bodies as a soluble protein. It should be noted that any endotoxins potentially existing as inadvertent contaminants in the N-Tc52/TSkb20 chimera were eliminated. Mass spectrometry analysis and SDS-PAGE of the purified recombinant protein revealed a molecular weight of 29.4 kDa, consistent with the predicted molecular mass of the chimera based on the sizes of the individual Tc52 and TS proteins previously reported [18]. The chimera N-Tc52/TSkb20 was identified by LC-MS/MS matching tryptic peptides resulting from both in-solution trypsin digestion of the purified protein and in-gel trypsin digestion of the protein band. Western blot analysis demonstrated that vaccination with N-Tc52/TSkb20 elicited a specific humoral immune response, as sera from immunized mice showed a single band of the correct molecular weight corresponding to the full-length recombinant protein. These findings confirmed that the construct maintained its anticipated structure and immunogenic epitopes following vaccination. These characterization techniques, including Western blot analysis, validated the approach by demonstrating accurate fusion of epitope sequences and characterization of the chimeric vaccine candidate. This served to verify protein production, purity, and identity prior to the evaluation of the protective efficacy of the vaccine.

### 4.2. Evaluation of N-Tc52/TSkb20 Protective Efficacy in a Mouse Model of Acute T. cruzi Infection

Previous studies have demonstrated the individual vaccine potential of both Tc52 and TS proteins from *T. cruzi* [14,16,17,22,56]. In the present study, we took advantage of the individual properties of these proteins and generated the N-Tc52/TSkb20 chimeric construct, which demonstrated that, when administered in naïve mice, it is capable of generating a protective response more efficiently in controlling parasite dissemination in the host compared to parental proteins and non-immunized mice. Notably, the tissue parasite burden was significantly reduced in the heart, colon and skeletal muscle of immunized mice, with the chimeric protein N-Tc52/TSkb20 eliciting stronger protection than the individual parental proteins rTc52 and rTS. While numerous amastigote nests were observed in tissues from non-immunized animals, classical intracellular parasitic nests were not detected in histological analyses of skeletal muscle from N-Tc52/TSkb20-immunized mice. This suggests that the chimeric protein rapidly and effectively controlled parasitemia with minimal inflammation and infection foci in the muscles, unlike the higher inflammation and foci seen with rTc52 and rTS immunization. In summary, the chimera N-Tc52/TSkb20 demonstrated enhanced protection and tissue clearance compared to individual protein components. Notably, N-Tc52/TSkb20 provided the highest level of protection following virulent challenge. It is worth noting that *T. cruzi* is a highly glycosylated microorganism [57,58,59], and that the presence of these glycosylations can positively influence immunogenic characteristics. Indeed, it is known that antibodies against glycoproteins exist in patients with acute and chronic Chagas disease targeting the linear α-Gal immunodominant glycotope. This glycotope elicits lytic and protective anti-α-Gal antibodies, which in turn control parasitemia in both the acute and chronic phases of the disease [60]. In this context, recombinant protein production in yeast emerges as a promising alternative production system due to its ability to perform post-translational modifications such as glycosylation. Yeasts naturally produce hypermannosylated glycosyl structures that can elicit immune responses [61]. Yeast are also easy to handle unicellular eukaryotes that grow rapidly to high cell density in cost-effective defined media, often with a high space-time yield. However, in this initial study, we were able to demonstrate the efficacy of a new antigen which, without requiring post-translational modifications, was able to significantly decrease infection by triggering a robust immune response. Interestingly, while the wild-type forms of Tc52, TS, and other parasitic proteins have been shown to intrinsically elicit immune responses elsewhere [14,17,62], the three proteins produced via prokaryotic expression in this study also demonstrated protective efficacy against acute *T. cruzi* infection. This outcome is consistent with studies showing protection in mice by other recombinant proteins also expressed in *E. coli*, such as the paraflagellar rod protein, cruzipain, TSA, Tc24, and several chimeric proteins such as Traspain [29,63,64,65]. Our findings highlight the immunological potential of rationally designed multi-epitope chimeras produced via bacterial systems, supporting prokaryotic expression as a valid platform for generating *T. cruzi* vaccine candidates. While post-translational modifications in native proteins may enhance the immune response, their absence was not found to be a determining factor based on the protective capacity induced by the N-Tc52/TSkb20 chimera.

### 4.3. Immunogenicity and Protective Mechanisms of N-Tc52/TSkb20 Vaccination

We sought to characterize the immune mechanisms underlying the protection afforded by N-Tc52/TSkb20 vaccination. While only modest antibody levels were detected overall, a notable finding was the detection of a predominance of IgG2c over IgG1 associated with enhanced control of intracellular pathogens [66,67]. In this prophylactic vaccination context, the predominant detection of IgG2c over IgG1 has tentatively been associated with an enhanced ability to control intracellular pathogens. Higher IgG2c levels compared to IgG1 could potentially contribute to decreasing *T. cruzi* parasite burden and improving infection management during both acute and chronic disease stages [68,69,70,71,72]. This raises the possibility that IgG2 predominance may play an important role in mitigating parasitemia and its progression throughout the lifecycle of this intracellular protozoan pathogen, presumably through augmented parasitic control [73,74].

In our study on cellular immunity, the AIM assay demonstrated that the TSkb20 peptide specifically activated CD8+ T-cells in N-Tc52/TSkb20-immunized mice compared to controls. The ELISPOT assay further showed higher IFNγ-secreting cells upon TSkb20 or whole N-Tc52/TSkb20 restimulation in N-Tc52/TSkb20 compared to non-immunized controls, demonstrating priming of this protective cytokine response. These findings align with those of previous studies, demonstrating the importance of establishing a TSkb20-specific CD8+ T-cell response to protect against *T. cruzi* infection. In one study, the TSkb20 epitope was loaded onto a bacteriophage, which showed a reduction in parasitemia, prolonged survival, and induction of TSkb20-specific CD8+ T-cell and antibody responses compared with controls in a C57BL/6 mouse host model [75]. Another study showed that expression of an immunodominant epitope in different viral vectors was sufficient to induce protection against the parasite through specific CD8+ cells in C57BL/6 mice [76,77]. The concordance between these corroborating observations reinforces the dominant role of the incorporated TSkb20 epitope in triggering a CD8+ response against *T. cruzi*. Our findings are in agreement with a recent study in which a single chimeric protein antigen was efficiently designed and expressed in *E. coli*. This protein was rationally engineered to present conserved CD8+ T-cell epitopes from the intracellular parasites, *Leishmania* spp. and *T. cruzi*. The recombinant chimeric protein was subsequently able to reduced parasite burdens in target organs after challenge with *L. infantum*, *L. mexicana* or *T. cruzi*, in comparison to control groups in female BALB/c mice [32]. The induction of effective CD8+ T-cell responses reinforces protective efficacy against intracellular parasitic challenges. The use of immunodominant epitopes as antigens in vaccines against Chagas disease is still under investigation. Previous studies have shown that the presence of immunodominant epitopes is not necessary for long-term protection, parasite loads in infected muscles during the acute phase of infection are higher in the absence of CD8+ T-cells specific for these epitopes [78]. Other researchers have also found that the response to subdominant epitopes provides some degree of protection in mice, but this does not compare to the protection obtained by immunodominant epitopes [79]. However, our results confirmed the importance of including dominant CD8+ T-cell epitopes to induce protective immunity, as was aimed at in our experimental vaccine approach using the chimeric antigen.

Given the critical role of T-cell immunity against *T. cruzi*, we also analyzed the CD4+ T-cell response induced by vaccination. Upon restimulation, our assays did not detect antigen-specific activation of the CD4+ T-cell population in response to the TSkb20 peptide. Our findings indicated that the vaccine primarily stimulated a CD8+ T-cell-mediated effector response rather than a CD4+ response. These results are consistent with those demonstrated by Biscari et al. (2022) [41], where an activation of CD8+ lymphocytes from C57BL/6 mice infected with *T. cruzi* upon re-stimulation with TSkb20 was observed, but induction of CD4+ T-cells was also not seen. Furthermore, splenocytes from all immunized groups secreted detectable levels of IL-10 upon stimulation. Notably, IL-10 levels were higher in the rTc52- and rTS-immunized groups than in the N-Tc52/TSkb20 group. This relatively low IL-10 environment in N-Tc52/TSkb20-immunized mice may favor antigen presentation and promote the innate immune response, which is critical for early parasite control. A reduced IL-10 background may enhance macrophage parasite elimination abilities and the capacity of dendritic cells to provide appropriate T-cell priming signals, constituting the first line of defense against *T. cruzi* [53,54]. This propitious immunological setting could improve the control of parasitemia in the initial stages of infection, thereby limiting tissue damage and reducing amastigote nest numbers in the skeletal muscles of N-Tc52/TSkb20-vaccinated animals compared with the rTc52 and rTS groups. Minimizing the parasitic load during the earliest phases of infection may be key to vaccine efficacy against intracellular pathogens, such as *T. cruzi*. While our primary objective was to study the efficacy of the chimeric antigen N-Tc52/TSkb20, when we tested the individual parental recombinant antigens, we found a moderate level of protection against *T. cruzi* challenge. This is consistent with previous studies that demonstrated the protective capabilities of both Tc52 and TS antigens individually. Early studies using the Tc52 protein as an immunogen demonstrated that it could confer partial protection when purified from parasites and formulated with adjuvant. DNA vaccination encoding Tc52 also promotes parasite clearance in BALB/c mice [14]. However, there are limitations to using antigens purified from parasites and the instability of naked DNA vaccines [80]. A more recent study analyzed the immune response and protection elicited by attenuated *Salmonella* by delivering a plasmid encoding full-length or domain fragments of Tc52 in C3H/HeN (H-2K) mice [21]. Early studies have shown that immunization with plasmid DNA encoding the catalytic domain of the TS protein induced a protective response against experimental *T. cruzi* infection in BALB/c mice, reducing peak parasitemia and mortality [81]. Subsequently, immunization with recombinant TS protein also elicited a protective response that protected 60% of the immunized mice when challenged with the parasite [82]. Moreover, a TS mutant lacking SAPA repeats (mTS) elicited stronger protective responses in mice than recombinant TS or non-vaccinated controls. Immunization with recombinant TS results in tissue inflammation and pathological changes upon challenge. The mTS design generated a suitable protective immune response profile. These results, among others previously presented, help illustrate the potential of the *T. cruzi* Tc52 and TS proteins, as vaccine antigens for Chagas disease. In our study, protection with Tc52 and TS was inferior to that of the chimeric antigen. We hypothesize that one possible cause is the formation of novel epitopes within the N-Tc52/TSkb20 chimeric sequence, which conferred stronger protection against *T. cruzi* challenge than the individual rTc52 and rTS proteins tested separately. Using an epitope prediction software, we identified novel peptides predicted to bind strongly to the MHC I class allele H2-Kb. Specifically regarding CD8+ T-cell responses, the NetMHCpan analysis revealed five peptides within the chimera that were predicted to bind strongly to the H2-Kb MHC class I allele in C57BL/6 mice and stimulate CD8+ T-cells. These potential CD8+ T-cell epitopes were derived from the fusion of sequences between N-Tc52 and two copies of TSkb20. Interestingly, they might share motifs with the known protective TSkb20 epitope, suggesting their potential to stimulate similar responses. In addition to stimulating CD8+ T-cell responses, peptides identified in the chimera using BepiPred identified a single novel B-cell epitope that was formed exclusively within the chimeric structure. This epitope does not correspond to sequences in individual proteins and may contribute to the qualitative and quantitative differences in immunogenicity between the chimera and its components, potentially including the more robust humoral response shown for N-Tc52/TSkb20, as demonstrated by the higher antibody levels. While experimental validation is needed, these in silico analyses indicate that the protective superiority of the N-Tc52/TSkb20 chimera over the individual antigens could stem from novel epitopes uniquely presented through its fused architecture. The identification of additional targeting epitopes for both CD8+ and B-cell responses could explain the enhanced immunity elicited by the chimeric vaccine. This highlights the potential of structure-based designs to alter epitope exposure and broaden immunological recognition. In this regard, modern approaches for the identification of *T. cruzi* epitopes, such as screening phage display libraries as previously described [83], could enable the identification of additional fragments to incorporate into our N-Tc52/TSkb20 protein to further improve an antigen against *T. cruzi* infection.

### 4.4. Significance and Future Study Directions

These initial studies successfully demonstrated that the novel N-Tc52/TSkb20 antigen produced in *E. coli* can protect against *T. cruzi* challenge while reducing parasitemia, tissue damage and nesting during acute infection. Our findings contribute to the growing body of literature tentatively associating certain vaccine-induced immune responses with partial protection against *T. cruzi* infection and disease progression. However, direct comparison with other subunit vaccines is limited by variability in preclinical models. Further investigation is needed to elucidate the protective mechanisms and fully characterize immunogenicity using different adjuvant platforms. Recent advances in adjuvant formulations and evaluations have expanded the application of *T. cruzi* vaccination strategies. Promising candidates studied in recent years have demonstrated enhanced immunogenicity when combined with recombinant antigens in prototype-subunit vaccine designs [15,29,84]. The incorporation of pathogen-associated molecular patterns as delivery systems is promising, with studies demonstrating the potential of bacterial components as carriers for priming robust immune responses [44,85]. Similarly, the use of virus-like particle technology carrying relevant antigens could exploit its ability to stimulate antigen presentation via MHC class I and II, thereby inducing balanced humoral and cellular immunity crucial for controlling intracellular parasites [86].

The next step is to evaluate the ability of the N-Tc52/TSkb20 vaccine to immunomodulate and reduce chronic Chagas damage. Future studies will concentrate on analyzing the parameters of cardiac damage, the primary manifestation in the chronic stage, such as the degree of fibrosis, levels of serological damage enzymes, electrocardiograms, and survival rates. Studies on long-term memory populations triggered by vaccination will also be assessed. Determining whether it can mitigate disease progression would support the development of this chimera as immunotherapy for chronic Chagas, the major public health challenge posed by this disease.

## 5. Conclusions

In summary, this study details the generation and characterization of the N-Tc52/TSkb20 chimera containing Tc52 and TSkb20 epitopes. Vaccination favorably modulated humoral and cellular immunity through the antibody, CD8+, and IFNγ pathways in the C57BL/6 mouse model, which expresses the H-2Kb MHC haplotype recognized by the TSkb20 epitope and is known to exhibit Th1 bias [87]. These findings demonstrate some of the protective mechanisms of action and the vaccine potential of this multi-epitope construct. However, limitations in the direct comparison between studies underscore the need for further optimization is still warranted. Continued development may establish this antigen as a clinical vaccine candidate by corroborating acute-phase protection and elucidating chronic disease prevention in various experimental settings. Overall, this proof-of-concept study supports the ongoing evaluation and refinement of this novel multi-epitope approach as a promising strategy for Chagas disease vaccination. With additional investigations, this platform shows potential for deployment against other intracellular pathogens.

## Figures and Tables

**Figure 1 vaccines-12-00621-f001:**
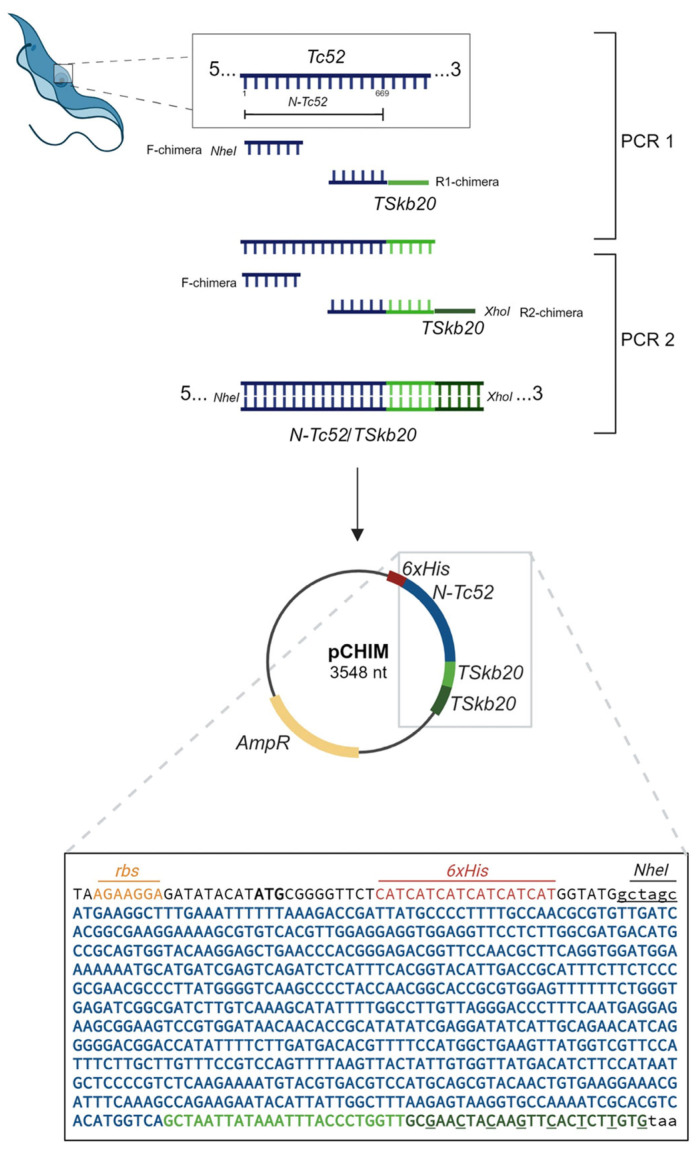
Molecular cloning of chimeric gene encoding N-Tc52/TSkb20 protein. The DNA sequence encoding the amino-terminal fragment of the Tc52 protein (represented in blue) was amplified and fused to two TSkb20-encoding sequences (in light and dark green). The resulting genetically linked sequence was cloned into the bacterial vector pRSET-A (Addgene), generating the plasmid p-CHIM (3548 nt). The inset shows the nucleotide sequence of the chimeric gene N-Tc52/TSkb20, which includes a 6x-His tag-encoding sequence from the pRSET-A vector (in red), the DNA encoding the amino terminal sequence of Tc52 (in blue), and two TSkb20-encoding sequences in tandem (in green). The ribosome binding site (RBS) is marked with an yellow overlay, and the underlined letters in the second TSkb20-encoding sequence represent modifications made in the primer R2-chimera to achieve the desired TSkb20 amino acid sequence by selectively choosing preferential *E. coli* codons while avoiding complementarity to the previously added sequence encoding the first TSkb20 epitope.

**Figure 2 vaccines-12-00621-f002:**
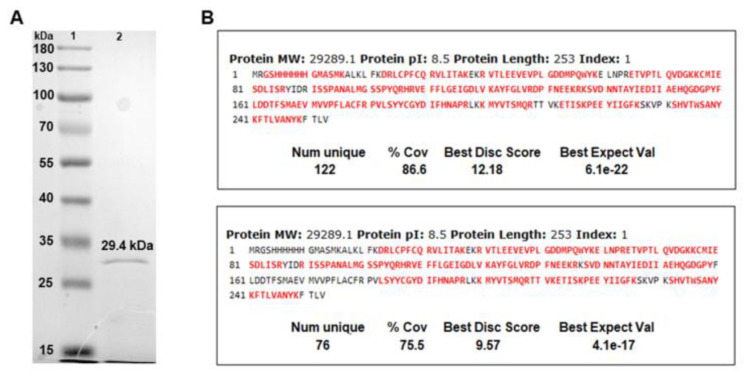
Characterization of N-Tc52/TSkb20. (**A**) Electrophoretic analysis of the pure chimeric protein in a 12% SDS-acrylamide gel: lane 1, molecular weight marker PageRuler Prestained Protein Ladder (Thermo Scientific); lane 2, N-Tc52/TSkb20 chimeric protein with a theoretical weight of 29.4 kDa. (**B**) Mass spectrometry-based peptide mapping of N-Tc52/TSkb20 after batch (**upper panel**) and in-gel digestion (**lower panel**). Mapped amino acid residues are highlighted in red.

**Figure 3 vaccines-12-00621-f003:**
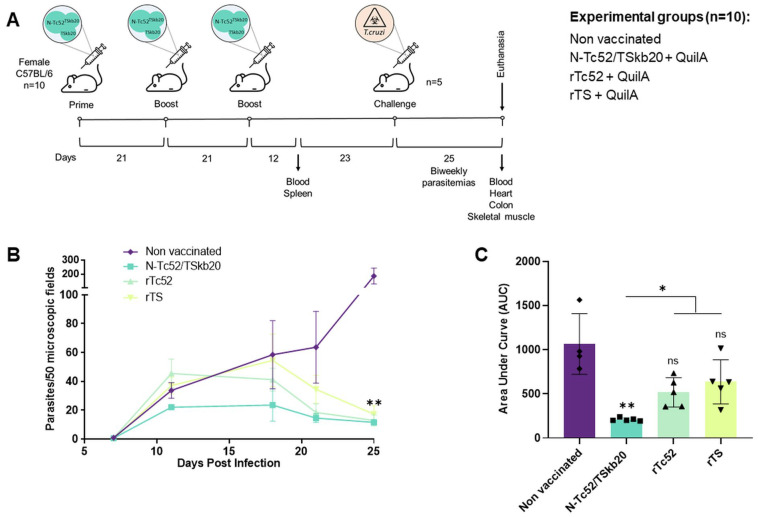
N-Tc52/TSkb20 immunization controls virulent *Trypanosoma cruzi* infection. (**A**) Immunization and challenge scheme. Female C57BL/6 mice were primed and boosted with N-Tc52/TSkb20 + QuilA, rTc52 + QuilA, rTS + QuilA, or QuilA alone (non-vaccinated control) at three-week intervals and subsequently challenged intraperitoneally with 1000 trypomastigotes from the Tulahuen strain 35 days after the final immunization. (**B**) Parasitemia curves during the acute phase of infection determined from 10 μL blood samples collected twice weekly. Parasites per 50 fields were enumerated microscopically (×40). (**C**) Area under the parasitemia curve (AUC) indicating the infection dynamics in each group. Data were analyzed using two-way ANOVA with Tukey’s multiple comparison test. Asterisks denote significance compared to the non-vaccinated group (** *p* < 0.01, * *p* < 0.05; ns, not significantly different).

**Figure 4 vaccines-12-00621-f004:**
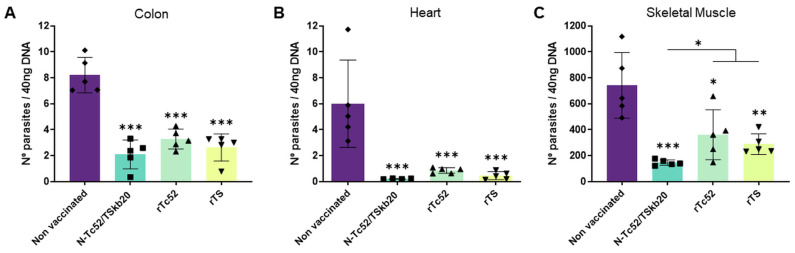
N-Tc52/TSkb20 immunization decreased parasite load in target organs following virulent *T. cruzi* infection. Quantification of parasite numbers by quantitative real-time PCR (qRT-PCR) amplification of the satellite DNA (Tc18Sr) sequence from (**A**) colon, (**B**) heart, and (**C**) skeletal muscle collected 25 d post-challenge. Data were analyzed using one-way ANOVA with multiple comparisons. Asterisks denote statistical significance compared to the non-vaccinated group (*** *p* < 0.001, ** *p* < 0.01, * *p* < 0.05) or between groups when specified.

**Figure 5 vaccines-12-00621-f005:**
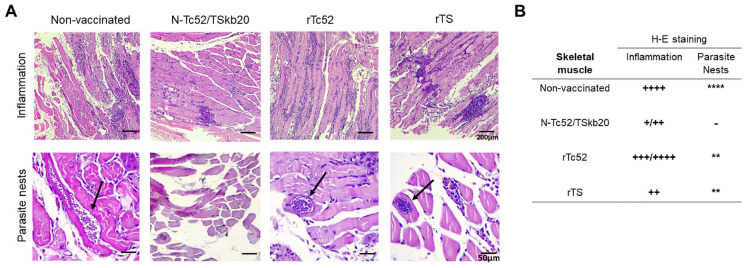
N-Tc52/TSkb20 immunization restricts amastigote nest presence and considerably reduces inflammation in skeletal muscle during acute *T. cruzi* infection. (**A**) Representative hematoxylin and eosin-stained skeletal muscle sections from all groups 25 d post-infection (dpi), imaged at 10× magnification. Black arrows indicate intracellular amastigote nests (examples shown circled at 40×). (**B**) All muscle sections were examined microscopically for intracellular parasites and inflammation. Amastigote nest numbers and inflammation were categorized as follows: Numbers of nests (- = absent, ** = 2–4, **** = 6–10). Inflammation was scored as + = small foci, ++ = medium foci, and +++/++++ = large foci.

**Figure 6 vaccines-12-00621-f006:**
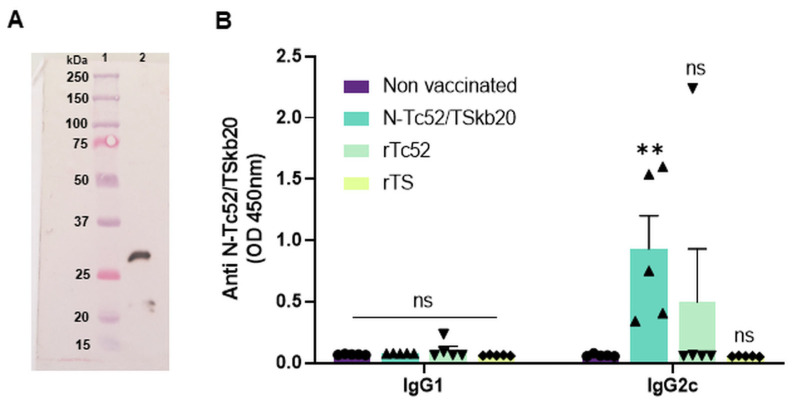
N-Tc52/TSkb20 vaccination elicits a specific antibody response. (**A**) Western blot confirmation of serum antibody-mediated detection of the N-Tc52/TSkb20 antigen. The chemiluminescent signal captured on photographic film overlays the transferred PVDF membrane, showing: Lane 1—Precision Plus Dual Color Prestained Protein Marker (Bio-Rad) for molecular weight estimation; Lane 2—1 μg of the N-Tc52/TSkb20 chimeric protein. (**B**) Serum samples collected 15 days after the third immunization from each experimental group were analyzed using ELISA for antibodies against the N-Tc52/TSkb20 chimera. IgG1 (**left**) and IgG2c (**right**) isotype levels induced in response to immunization. Serum samples were incubated in plates coated with N-Tc52/TSkb20 antigen to determine antibody titers. The N-Tc52/TSkb20 vaccination elicited higher IgG2c antibody levels than the other immunization groups. Asterisks denote statistical significance compared to the non-vaccinated group (** *p* < 0.01; ns, not significantly different).

**Figure 7 vaccines-12-00621-f007:**
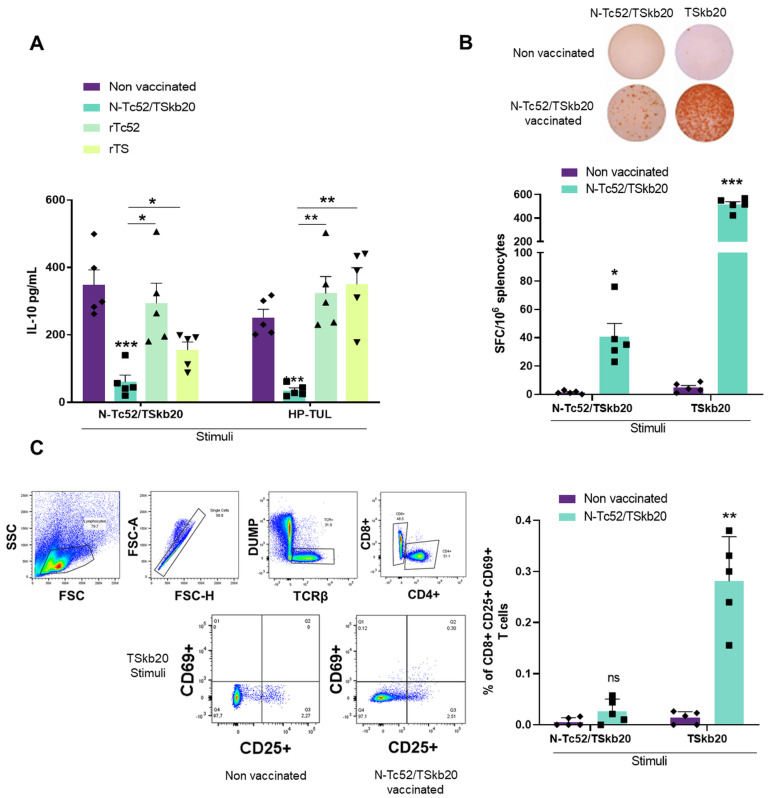
N-Tc52/TSkb20 vaccination decreased IL-10 levels and elicited antigen-specific cellular immune responses. (**A**) IL-10 levels were determined by ELISA in the supernatants of total splenocytes stimulated for 48 h. (**B**) Representative ELISPOT images (**top**) and quantification of IFNγ spot forming cells in spleen after restimulation with TSkb20 or N-Tc52/TSkb20 (**bottom**). (**C**) N-Tc52/TSkb20 vaccination elicited antigen-specific CD8+ immune responses. Flow cytometry assay showing the gating strategy used to identify CD8+ CD69+ CD25+ T-cells in the spleen (**left**) and frequencies of these cells after restimulation with TSkb20 peptide or N-Tc52/TSkb20 protein (**right**). Data were analyzed using two-way ANOVA with Tukey’s multiple comparison test. Asterisks denote statistical significance compared to the non-vaccinated group (*** *p* < 0.001, ** *p* < 0.01, * *p* < 0.05; ns, not significantly different) or between groups when specified.

**Table 1 vaccines-12-00621-t001:** Oligonucleotides used in this study.

	Nucleotide Sequences (5′-3′)
F-chimera	CTAGCTAGCATGAAGGCTTTGAAACTTTTTAAAG
R1-chimera	GTTCGCAACCAGGGTAAATTTATAATTAGCTGACCATGTGACGTGC
R2-chimera	CCGCTCGAGTTACACAAGAGTGAACTTGTAGTTCGCAACCAGGGTAAAT
F-TS	CTAGCTAGCAGAAGGTCAATGGGAAAG
R-TS	CCGCTCGAGTCACAGAGCCGCAAACCCCCAC
F-Tc52	GGACTGCAGGAAGGCTTTGAAACTTTT
R-Tc52	GGAAAGCTTTCAAGACGATGGACGCAAA
F-Sat	GCAGTCGGCKGATCGTTTTCG
R-Sat	TTCAGRGTTGTTTGGTGTCCAGTG

Sequences of oligonucleotides used in the construction of expression plasmids pCHIM, pTS, and pTc52 from which the recombinant proteins N-Tc52/TSkb20, rTS, and rTc52 were obtained, as well as oligonucleotides used for qRT-PCR detection of parasite DNA.

**Table 2 vaccines-12-00621-t002:** Predicted CD8+ T- and B-cell epitopes.

Amino Acid Sequences	Epitope Characteristics
Binding MHCI epitopes	Type of binding/allele
SANYKFTL	SB H2-K^b^ (New)
SANYKFTLV	WB H2-D^b^ and SB H2-K^b^ (New)
ANYKFTLV	SB H2-K^b^ (TSkb20)
ANYKFTLVA	SB H2-K^b^ (New)
VANYKFTL	SB H2-K^b^ (New)
VANYKFTLV	SB H2-K^b^ (New)
B-cell epitopes	Origin
SHHHHHHGMASMK	Histidine tag
EVPLGDDMPQWYKELNPRE	Tc52
RDPFNEEKRKSVDN	Tc52
YCGYDIFHNAP	Tc52
RTTVKETISKPE	Tc52
PKSHVTWSANYK	New

This table shows the amino acid sequences of potential epitopes predicted by the NetMHCpan-4.1 and IEDB software tools. CD8+ T-cell epitopes indicate novel sequences identified as forming through the fusion of the two TSkb20 sequences in the chimeric vaccine. Binding strengths of H2-Kb and H2-Db alleles in C57BL/6 mice are shown. SB = Strong Binding; WB = Weak Binding. For B-cell epitopes, sequences mapping to the N-Tc52/TSkb20 protein component are listed.

## Data Availability

The data that support the findings of this study are available from the corresponding author, upon reasonable request.

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
