# Peer review of "Protective Efficacy of the Epitope-Conjugated Antigen N-Tc52/TSkb20 in Mitigating Trypanosoma cruzi Infection through CD8+ T-Cells and IFNγ Responses"

_vaccines, 2024, doi:10.3390/vaccines12060621_

Round 1

Reviewer 1 Report

Comments and Suggestions for Authors

The manuscript by Vázquez and colleagues demonstrates the advantage of using a biotech-produced chimeric antigen composed of conserved epitopes from T. cruzi antigens. The authors described the design, production and purification of the chimeric vaccine and demonstrated improved immune protection in murine model (in the reduction of parasite load, production of interferon and stimulation of CD8+ response) when compared to parental/WT antigens inoculation.

The study is of medical and biological relevance and the manuscript is very well written, with logic designed experiments and clear results. I have some minor comments, a couple of suggestions and questions for the authors that should be addressed before accepting for publication. 

Minor comments/corrections

250: excessive space between words “consisting___of 50 µL”.

269 (and elsewhere): please verify the official nomenclature for the Tulahuen strain (I only know the term without the stress sign in “e”).

357: in the “1x106” the number 6 is not superscripted.       

616: Figure 8A is mentioned, but it does not exist (perhaps 7 C).

Suggestions

1)    Title: In my opinion, the manuscript title is generic for an experimental paper, authors could consider using a more “direct to the point”/specific title mentioning the antigen composition and/or the major immune effects, in line with the proposed titles from references #29 or #33, for example.

2)   Images: resolution must be improved, particularly for Figure 1 the resolution is very poor, and some written content is very difficult to read.

3)    I think the authors could check if some content from Results section 3.1 is not repetitive from Methodology sections 2.1 - 2.3. I observed the same for the Figure 1 legend. Since the design and production of the chimeric antigen are also part of the results, the methodological parts should be shortened (or simply referred to), to focus directly on results obtained.

4)To facilitate visualization, Figure 4 graphs could have titles with the evaluated organs/tissues

5)    The authors did not show a WB of parental proteins rTc52 and rTS, which were produced and purified similarly to chimeric N-Tc52/TSkb20, perhaps adding as supplem. information would be relevant for verification.

Questions

1)    T. cruzi strains are commonly associated with distinct disease profiles, tissue tropism and variable response to chemotherapy and immunity. Why did the authors decide to work with CL Brener as model for antigen development and what was the rational to use Tulahuen strain for the murine infection model? A recent paper (10.1080/22221751.2024.2315964) describes that approx. 10% of T. cruzi proteome is highly conserved among the distinct DTUs, the authors also mentioned that Tc52 is conserved among strains, how about TSkb20, is the fragment also conserved towards distinct DTU?

2)     Since the chimeric antigen, a combination of 2 fragments from rTc52 and rTS, is systematically compared to separated effects of parental antigens, would it make sense to test the parent fragments in combination for comparison, in a dual-antigenic fashion? Do you expect to have a similar effect to the chimeric antigen?

3)    463: animal survival recordings are mentioned, why a survival curve was not displayed if a lethal load of parasite was used in the mice?

4) 790: It is not clear why the use of immunodominant epitopes is controversial in CD immunization. Please elaborate more on this aspect in the Discussion.

Author Response

Please find thereafter a point-by-point response to the reviewers’ queries. 

Reviewer 1:

The manuscript by Vázquez and colleagues demonstrates the advantage of using a biotech-produced chimeric antigen composed of conserved epitopes from T. cruzi antigens. The authors described the design, production and purification of the chimeric vaccine and demonstrated improved immune protection in murine model (in the reduction of parasite load, production of interferon and stimulation of CD8+ response) when compared to parental/WT antigens inoculation.

The study is of medical and biological relevance and the manuscript is very well written, with logic designed experiments and clear results. I have some minor comments, a couple of suggestions and questions for the authors that should be addressed before accepting for publication. 

We appreciate and thank the reviewer for their valuable feedback. The manuscript has been revised to reflect their suggestions. Please find thereafter a point-by-point response to the queries. 

Minor comments/corrections

250: excessive space between words “consisting___of 50 µL”.

This was corrected (line 256).

269 (and elsewhere): please verify the official nomenclature for the Tulahuen strain (I only know the term without the stress sign in “e”).

Thank you for catching that mistake. This error has been corrected throughout the manuscript.

357: in the “1x106” the number 6 is not superscripted.   

This was corrected.

616: Figure 8A is mentioned, but it does not exist (perhaps 7 C).

Thank you for catching that mistake. The reference to the figure was corrected.

Suggestions

  • Title: In my opinion, the manuscript title is generic for an experimental paper, authors could consider using a more “direct to the point”/specific title mentioning the antigen composition and/or the major immune effects, in line with the proposed titles from references #29 or #33, for example.

Thank you for the feedback. The title has been changed to "Protective efficacy of the epitope-conjugated antigen N-Tc52/TSkb20 in mitigating Trypanosoma cruzi infection through CD8+ T-cells and IFNγ responses".

2)   Images: resolution must be improved, particularly for Figure 1 the resolution is very poor, and some written content is very difficult to read.

The quality of the figures has been improved. All of them now have a resolution of 3000 dpi, which exceeds the minimum quality indicated in the submission guidelines. Please note that PDF creation by the editorial software could impact figure quality.

3)    I think the authors could check if some content from Results section 3.1 is not repetitive from Methodology sections 2.1 - 2.3. I observed the same for the Figure 1 legend. Since the design and production of the chimeric antigen are also part of the results, the methodological parts should be shortened (or simply referred to), to focus directly on results obtained.

Thank you for the suggestion. We reviewed and modified section 3.1 of the Results to focus on the findings obtained, avoiding repeating methodological details described in sections 2.1 to 2.3. Likewise, we briefly simplified the legend of Figure 1 to avoid duplicating aspects already clearly explained in the Methods while maintaining clarity of the figure explanation. With these modifications, we believe we have optimized the presentation by focusing directly on communicating the results obtained, without repeating methodological details, as the reviewer appropriately proposed.

4)To facilitate visualization, Figure 4 graphs could have titles with the evaluated organs/tissues

Figure 4 was modified according this suggestion.

5)    The authors did not show a WB of parental proteins rTc52 and rTS, which were produced and purified similarly to chimeric N-Tc52/TSkb20, perhaps adding as supplem. information would be relevant for verification.

Thank you for the suggestion. A Supplementary figure (Figure S1) was added showing the identification of the pure rTc52 and rTS proteins in polyacrylamide gels. Reference to this figure was added in the manuscript in Section 2.2 (lines 201-202).

Questions

1)    T. cruzi strains are commonly associated with distinct disease profiles, tissue tropism and variable response to chemotherapy and immunity. Why did the authors decide to work with CL Brener as model for antigen development and what was the rational to use Tulahuen strain for the murine infection model? A recent paper (10.1080/22221751.2024.2315964) describes that approx. 10% of T. cruzi proteome is highly conserved among the distinct DTUs, the authors also mentioned that Tc52 is conserved among strains, how about TSkb20, is the fragment also conserved towards distinct DTU?

We appreciate this feedback. It's an interesting question that warrants further discussion. The TSKB20-specific CD8+ T-cell response in T. cruzi infection is known to be one of the strongest responses against a single epitope in any infectious disease. The variation in immunodominance of CD8+ T-cell responses to T. cruzi depending on the infecting strain of the parasite suggests differences in expressed ts genes in different strains (DOI: 10.1371/journal.ppat.0020077). This can lead to variability in the immunodominant targets of the CD8+ T-cell response among different parasite strains. Anyway, TSkb20 elicits CD8+ T-cell responses that show significant conservation across different DTUs. Studies have shown that the TSkb20 epitope can stimulate T cells from infections with different T. cruzi DTUs, indicating sequence conservation of this epitope among the major parasite lineages (doi.org/10.1371/journal.pntd.000914). This observation is now mentioned in the Introduction section, lines 84-87. Otherwise, taking into account that the tc52 gene is well conserved among distinct T. cruzi DTUs and that the TSKB20 epitope was incorporated to the fusion protein by successive PCRs, it was practically irrelevant what DTU background was used for antigen development. However, for the murine model we wished to ensure the use of a T. cruzi strain that reliably produces clear and noticeable parasitemia, as well as a mouse immunization and challenge model that we are well acquainted with. The Tulahuen strain has been maintained in our lab for more than 20 years through in vivo passage, and through this long experience we have gained deep familiarity with its behavior and can rely on well-established standardized murine models of infection.

2)     Since the chimeric antigen, a combination of 2 fragments from rTc52 and rTS, is systematically compared to separated effects of parental antigens, would it make sense to test the parent fragments in combination for comparison, in a dual-antigenic fashion? Do you expect to have a similar effect to the chimeric antigen?

This is an interesting question. The current work compares the novel chimeric protein to its parental proteins in terms of protection against T. cruzi infection. It is true that using the proteins together may have similar effects, and their evaluation could be merited. However, combining distinct epitopes in a single molecule conveys certain advantages that demonstrate our choice in developing this vaccine, such as: i) Immunological: the combination of epitopes seeks generate simultaneous immune responses to several epitopes, increasing magnitude and duration of protection conferred. In our case, we even found formation of new sequences that could act as epitopes. The presence of varied domains in the fusion protein increases the likelihood of stimulating immune recognition in diverse strains or individuals as a single protein must be captured and processed by antigen presenting cells; ii) Practical: including multiple protein epitopes in a single protein greatly simplifies vaccine application by requiring purification of only one protein and administration of a single molecule instead of multiple proteins; iii) Size: the molecular size of the fusion protein is relatively small, smaller than the sum of the individual proteins, facilitating recombinant production and purification, especially in a prokaryotic model like the one used in our work; iv) Economic: reducing complexity and component number decreases production, manufacturing, storage, and distribution costs for the final vaccine and, v) Experimental: allows simultaneous evaluation of immunogenicity and protective efficacy of multiple antigens in a single preclinical model, streamlining experiments, approvals by ethics committees and regulatory agencies.

3)    463: animal survival recordings are mentioned, why a survival curve was not displayed if a lethal load of parasite was used in the mice?

Thank you for the feedback. We appreciate the observation. The endpoint was established upon the first animal death and evident deterioration of health in the remaining animals. In this work context, we did not show a survival graph as this graph would not provide much insight since all animals were humanely euthanized after the death of the first animal. This was done to enable comparison of the degree of infection, distribution in organs, parasitic load and inflammation in tissues at the same time point for each condition. To provide further clarity, we have amended the phrasing "lethal load of parasite" to "infective load of parasite" in lines 270, 275 and 462.

4) 790: It is not clear why the use of immunodominant epitopes is controversial in CD immunization. Please elaborate more on this aspect in the Discussion.

Thank you for the suggestion. We agree that further elaboration on this topic would help to improve clarity and discussion of this important issue. We have clarified this point by describing it in more detail in lines 802-810 of Discussion Section. References have been added. Additionally, the word "controversial" was changed to soften the assessment.

Reviewer 2 Report

Comments and Suggestions for Authors

It is an interesting paper which expands on the current knowledge of vaccine development against Chagas disease. Briefly, this study addresses the development of a vaccine candidate against T. cruzy, conducted in an in vivo murine model and assessing vaccine efficacy with several parameters further analyzed, such as parasite load, tissue damage and elicited immune response. It is worthy study that adds insights on the potential control of the disease.

I consider that the work is well written, the methodological procedures were thoroughly detailed, the experimental design is correct,  and  results clearly expressed and discussed. However, I believe there is place for improvement. Please, do accept my suggestions as follows:

Line 110 tp 122: I suggest authors to eliminate this paragraph from the introduction section since it includes results and assessment of key findings. Results/discussion should not be given in advance in this section .

Line 303: IgG subtypes (on onwards): replace subtypes by subclasses.

What coating buffer was used to coat ELISA plates? Same inquire for point 2.11 (IL-10) 

Line 303. Please do add (at the end of point 2.8), how results (antibody production) were finally expressed

Line 391, point 2.15: Although it is well known, please do add what P-values were considered as statistically significant?

Line 410: “in vivo” is not in italic

I suggest avoiding assessment of results within results section, since in discussion section is were the actual evaluation and assessment should be done. Therefore, I suggest eliminating words suchs as Remarkably (line 489), succesfuly (519), or sentences were assessment was conducted (502-504), “consistent with our previous results” (508); from the results section. Also, line 517, there is assessment of results.

There are other sections within “results” that include assessment/discussion of results. (point 3.3, line 531, point 3.4), (also 570-573; 637-645). Overall, I suggest authors to partly rewrite/reorganise and revise the “results” section.  I consider that some points of the results  could be summarized, basically restricting to a detailed description of the result and avoiding providing an assessment and discussion. Summarizing the results and highlighting  the key findings would make the reading experience of this manuscript more straightforward to the reader.

Discussion: results were evaluated in detail. However, I do believe the discussion section can also be summarized, some reading points were repetitive considering the above-mentioned comments.

Some of the results were consistent with other studies (comparative assessment). While discussing, I suggest mentioning the animal host where those in vivo studies were carried out.

Author Response

Reviewer 2:

It is an interesting paper which expands on the current knowledge of vaccine development against Chagas disease. Briefly, this study addresses the development of a vaccine candidate against T. cruzy, conducted in an in vivo murine model and assessing vaccine efficacy with several parameters further analyzed, such as parasite load, tissue damage and elicited immune response. It is worthy study that adds insights on the potential control of the disease.

I consider that the work is well written, the methodological procedures were thoroughly detailed, the experimental design is correct, and results clearly expressed and discussed. However, I believe there is place for improvement. Please, do accept my suggestions as follows:

We appreciate and thank the reviewer for their valuable feedback. The manuscript has been revised to reflect their suggestions. Please find thereafter a point-by-point response to the queries.

Line 110 tp 122: I suggest authors to eliminate this paragraph from the introduction section since it includes results and assessment of key findings. Results/discussion should not be given in advance in this section .

Thank you for the feedback, the paragraph has been removed as suggested by the reviewer.

Line 303: IgG subtypes (on onwards): replace subtypes by subclasses.

The term "subtypes" has been replaced with "subclasses" throughout the manuscript.

What coating buffer was used to coat ELISA plates? Same inquire for point 2.11 (IL-10) 

We apologize for this oversight. The coating buffer has been added to Sections 2.8 and 2.11.

Line 303. Please do add (at the end of point 2.8), how results (antibody production) were finally expressed

This information has now been added at the end of Section 2.8.

Line 391, point 2.15: Although it is well known, please do add what P-values were considered as statistically significant?

This information has now been added at the end of Section 2.15.

Line 410: “in vivo” is not in italic

This was corrected.

I suggest avoiding assessment of results within results section, since in discussion section is were the actual evaluation and assessment should be done. Therefore, I suggest eliminating words suchs as Remarkably (line 489), succesfuly (519), or sentences were assessment was conducted (502-504), “consistent with our previous results” (508); from the results section. Also, line 517, there is assessment of results.

Thank you for this valuable feedback. We have carefully considered the observation and have updated relevant sections of the Results according to the reviewer's suggestions. See lines 406-423, 427-428, 485, 498, 511-513, 517-518, 526-528, 543-544, 581-584, 648-656, 679-681.

There are other sections within “results” that include assessment/discussion of results. (point 3.3, line 531, point 3.4), (also 570-573; 637-645). Overall, I suggest authors to partly rewrite/reorganise and revise the “results” section.  I consider that some points of the results  could be summarized, basically restricting to a detailed description of the result and avoiding providing an assessment and discussion. Summarizing the results and highlighting  the key findings would make the reading experience of this manuscript more straightforward to the reader.

Discussion: results were evaluated in detail. However, I do believe the discussion section can also be summarized, some reading points were repetitive considering the above-mentioned comments.

Thank you for the feedback. We have carefully considered the point raised by the reviewer and made the appropriate modifications, both in the specified lines detailed for the reviewer and other relevant sections of Results. A similar approach was taken for some parts of Discussion as well. The original aim was to avoid providing assessment and discussion within the Results Section. In Discussion, we streamlined repetitive ideas to enhance readability and remove unnecessary repetition, while maintaining the main points.

Some of the results were consistent with other studies (comparative assessment). While discussing, I suggest mentioning the animal host where those in vivo studies were carried out

We appreciate this reviewer's suggestion. We have now mentioned the animal host in the comparative assessment of in vivo studies in line 791 and throughout the 4.3 Discussion Section.